A retrospective study of automatic progressive optimization for lung cancer radiotherapy plans on the Halcyon and RayStation systems

Shao Kainan 1
Du Fenglei 2
Qiu Lingyun 1
Zhang Yinghao 1
Li Yucheng 1
Ding Jieni 1
Zhan Wenming 1
Chen Weijun chenweijun@hmc.edu.cn 1
1 Department of Radiation Oncology, Zhejiang Provincial People’s Hospital , Hangzhou , Zhejiang , China
2 Department of Radiation Physics, Zhejiang Cancer Hospital , Hangzhou , Zhejiang , China
Patnaik Santosh
Electronic publication date: 2025 Aug 5
Publication date: 2025
Volume: 13
Electronic Location ID: e19831
Received 2025 Mar 12; Accepted 2025 Jul 11
Copyright: ©2025 Shao et al.
Copyright year: 2025
Copyright holder: Shao et al.
License: This is an open access article distributed under the terms of the Creative Commons Attribution License, which permits unrestricted use, distribution, reproduction and adaptation in any medium and for any purpose provided that it is properly attributed. For attribution, the original author(s), title, publication source (PeerJ) and either DOI or URL of the article must be cited.
License URL: https://creativecommons.org/licenses/by/4.0/

Keywords: Automated treatment planning, Halcyon accelerator, Volumetric modulated Arc therapy, Lung cancer radiotherapy

Funding: Zhejiang Province Natural Science Foundation of China LGF21H180014 Medical and Health Research Project of Zhejiang Province 2021PY002 Zhejiang Provincial Basic Public Welfare Research Project LGF22H160070 This research was supported by the Zhejiang Province Natural Science Foundation of China under Grant No. LGF21H180014, the Medical and Health Research Project of Zhejiang Province under Grant No. 2021PY002, and Zhejiang Provincial Basic Public Welfare Research Project (No. LGF22H160070). There was no additional external funding received for this study. The funders had no role in study design, data collection and analysis, decision to publish, or preparation of the manuscript.

==============================
This study aims to evaluate the feasibility of using RayStation’s scripting function to generate automated radiotherapy plans for non-small cell lung cancer (NSCLC) patients on a Varian Halcyon accelerator and to compare their dosimetric characteristics with those of retrospectively collected manual clinical plans. A total of 63 conventional fractionation plans for NSCLC, previously designed using RayStation 4.5 for a variety of linear accelerators—including Trilogy, TrueBeam, Halcyon, and Elekta Infinity—were compared with automated plans generated using RayStation 9.0 for Halcyon. This heterogeneous control group was chosen to reflect real-world clinical practice across multiple platforms. Target coverage, doses to organs at risk (OARs), monitor units, and plan complexity were assessed. The automated plans showed improved dose conformity and lower OAR exposure under the planning configuration used. However, these differences should be interpreted with caution, as the comparison involved different treatment planning systems (TPS) versions and hardware platforms. Further controlled studies using the same TPS and linac are needed to validate the observed improvements.

Introduction

Non-small cell lung cancer (NSCLC) is one of the leading causes of cancer-related death globally and accounts for 85% of all lung cancer cases (Molina et al., 2008). Although advancements in treatment technology have improved patient outcomes, the overall survival rate for patients with NSCLC remains low, making the optimization of treatment methods crucial. Volumetric modulated arc therapy (VMAT) has become an essential technique in NSCLC radiotherapy due to its favorable dose distribution, reduced treatment time, and lower exposure to surrounding healthy tissues. According to the 2023 version of the National Comprehensive Cancer Network (NCCN) guidelines (Ettinger et al., 2023), VMAT is recommended for definitive and palliative thoracic radiotherapy in NSCLC. Recent studies have also validated its performance on modern treatment platforms, including the Halcyon linear accelerator (Huang & Liu, 2023). Radiation pneumonitis (RP), a common complication following NSCLC radiotherapy administration, is closely linked to the dose distribution of the radiotherapy plan (Rodrigues et al., 2004). Studies have suggested that reducing the volumes receiving at least 20 Gy (V20Gy) (Graham et al., 1999; Tsujino et al., 2003), 30 Gy (V30Gy) (Piotrowski, Matecka-Nowak & Milecki, 2005; Hernando et al., 2001), and 5 Gy (V5Gy) (Wang et al., 2006), and the mean lung dose (MLD) (Kwa et al., 1998; Yorke et al., 2002) can decrease the incidence of RP. Hence, optimizing radiotherapy plans to minimize the dose to normal lung tissue is of great clinical importance. However, owing to variability in target location and size, as well as patient-specific differences, traditional manual plan designs and evaluations are limited. Some plans, although meeting clinical protocol requirements and obtaining approval from clinicians, still have opportunities for further optimization, particularly in reducing the dose to normal tissues, especially the lungs.

In clinical practice, radiotherapy plan formulation typically involves multiple iterations and complex adjustments, which are time-consuming and labor-intensive and require significant expertise from physicists and dosimetrists. Following clinician confirmation of the treatment target volumes and organs at risk, treatment plan formulation involves steps such as creating nonanatomical auxiliary structures, selecting treatment techniques, arranging irradiation fields, setting target parameters, repeatedly optimizing the plan, and adjusting irradiation fields and parameters according to the optimization results. The differences in experience levels among the professionals involved in the plan design process can also lead to dose discrepancies, affecting treatment outcomes. To improve the efficiency and consistency of radiotherapy plan generation, automated planning techniques such as knowledge-based planning (KBP) (Zhu et al., 2011; Appenzoller et al., 2012), multicriteria optimization (MCO) (Thieke et al., 2007; Halabi, Craft & Bortfeld, 2006; Craft & Bortfeld, 2008), and scripting-based automatic treatment planning (ATP) have been developed (Nguyen et al., 2022). The RayStation treatment planning system supports scripting through Python interfaces and has been successfully applied in auto-planning workflows across various tumor sites (Pallotta et al., 2021; Lou et al., 2023; Yedekci et al., 2023). These systems allow for adaptive control over plan objectives, dose gradients, and complexity, enabling reproducible and clinically acceptable plans across institutions. Radiotherapy plans can also be designed and optimized in treatment planning systems (TPSs), such as Pinnacle (Philips Radiation Oncology Systems, Fitchburg, WI) (Gintz et al., 2016), Eclipse (Varian Medical Systems, Palo Alto, CA, USA) (Hussein et al., 2018; Ge & Wu, 2019), Monaco (Elekta AB, Stockholm, Sweden) (Bijman et al., 2021; Biston et al., 2021), and RayStation (RaySearch Medical Laboratories AB, Stockholm, Sweden) (Bodensteiner, 2018; Kierkels et al., 2015), each presenting with unique characteristics in terms of plan optimization and dose calculation algorithm. Among them, the direct machine parameter optimization (DMPO) algorithm used in the RayStation planning system yields particularly high efficiency and accuracy in terms of radiotherapy plan design (Hårdemark et al., 2003). The advantage of the DMPO algorithm lies in its direct iterative optimization of physical parameters on the basis of the current dose distribution, simplifying the optimization process and reducing potential errors during dose-to-parameter conversion, resulting in the development of more precise radiotherapy plans (Dobler et al., 2007; Dobler et al., 2009). Previously, we used IronPython in the RayStation TPS to design automated radiotherapy plans for nasopharyngeal carcinoma, validating the clinical applicability of this system. The dose distribution in the automated plans (APs) for tumor targets and organs at risk was similar or superior to that in manual plans (MPs), significantly improving clinical efficiency by reducing manual operation time (Yang et al., 2020).

Furthermore, the quality and efficiency of treatment plans are closely tied to the capabilities of the delivery platform. The Halcyon accelerator, a next-generation O-ring linac, provides several hardware features—such as dual-layer multileaf collimator (MLC) and high-speed gantry rotation—that facilitate efficient implementation of advanced automated planning strategies. The Halcyon accelerator features a jaw-free gantry design and dual-layer stacked MLC with 29 pairs of leaves on the proximal end and 28 pairs on the distal end, a maximum field size of 28 cm × 28 cm, six megavoltage (MV) X-rays in flattening filter-free (6MV-FFF) beam energy, a maximum gantry rotation speed of 24 °/s, a maximum leaf motion speed of five cm/s, and an effective leaf width of 0.5 cm (Lim et al., 2019; Roover et al., 2019; Riley et al., 2018). These characteristics allow the Halcyon accelerator to achieve excellent results in delivering radiotherapy for head and neck (Michiels et al., 2018), thoracic (Flores-Martinez et al., 2019; Huang & Liu, 2023), abdominal, and gynecological tumors (Jarema & Aland, 2019; Li et al., 2019), with good dose distribution, plan quality, and gamma pass rates (Laugeman et al., 2020). Additionally, the Halcyon offers substantial benefits in performing image-guided radiation therapy (IGRT), providing faster and more accurate imaging, thus improving treatment accuracy and efficiency (Hermida-López et al., 2023). The RayStation planning system facilitates the convenient formulation and optimization of treatment plans for the Halcyon accelerator (Saini et al., 2021). Therefore, in this study, we chose to use RayStation’s scripting functionality (Eley, 2016) and the Halcyon accelerator model to automatically design and optimize NSCLC radiotherapy plans. We compared the automated plans with retrospectively collected manual plans in terms of target coverage, OAR doses, MU usage, and plan complexity. As the two groups were generated using different TPS versions and linac platforms, the results reflect the performance under the current configuration and require further validation under matched conditions.

Materials and Methods

Patient selection

Data collection and inclusion criteria were consistent with our previous methodology (Shao et al., 2025). Specifically, this retrospective study included 63 patients who received radiotherapy for NSCLC from 2021 to 2023, all prescribed 60 Gy in 30 fractions. Eligible patients had histologically confirmed NSCLC, underwent routine radiotherapy during the specified period, and had complete computed tomography (CT) imaging data available for treatment planning.

Patients were excluded if they had previously undergone thoracic radiotherapy, did not receive the prescribed dose of 60 Gy in 30 fractions, or had incomplete treatment records. This retrospective study was approved by the Medical Ethics Committee of Zhejiang Provincial People’s Hospital (No. QT2024085) and complied with the Declaration of Helsinki. Patient consent was waived due to the anonymized nature of the data.

The comparison in this study involved automated plans generated for the Halcyon platform and previously implemented clinical plans. The control group comprised historical treatment plans designed using a variety of clinical linac platforms, including Trilogy, TrueBeam, Halcyon, and Elekta Infinity. This heterogeneity arose from the retrospective nature of the clinical control group, which included treatment plans delivered on multiple linear accelerator platforms. Therefore, the comparisons should be interpreted with consideration of platform-related differences.

Simulation positioning

The CT simulation procedures also followed previously reported methods (Shao et al., 2025). In brief, existing CT images acquired during routine radiotherapy of the 63 NSCLC patients were used. Simulations were conducted using a Brilliance Big Bore CT simulator (Philips, Netherlands). Patients were positioned supine with arms raised above the head and immobilized using a thermoplastic mask and positioning board.

The scan covered the region from the upper edge of the second cervical vertebra to the lower edge of the second lumbar vertebra, with a slice thickness of five mm. The images were exported in DICOM format and imported into RayStation version 9A for subsequent planning and analysis.

Target and organs-at-risk delineation

The target volumes and organs at risk (OARs) were delineated by experienced physicians. The target volumes included the primary lung tumor (GTV-T) and positive lymph nodes (GTV-N), forming the total gross tumor volume (GTV). The clinical target volume (CTV) consisted of the extended GTV-T region (CTV-T) and the entire positive lymph node area (CTV-N). The planning target volume (PTV) was defined by expanding the CTV by five mm and was reviewed and modified by the physicians. The OARs included both lungs, the heart, and the spinal cord.

Treatment planning

The manual planning control group consisted of retrospectively selected radiotherapy plans designed manually by experienced physicists and evaluated by physicians and created on a RayStation planning system (versions 4.5 and 9A). The inclusion criteria for the patients on whom the plans were implemented included conventional fractionated lung cancer radiotherapy between 2021 and 2023 with a prescription dose of 60 Gy/30F. The accelerators used included Varian’s Trilogy (39 patients, 61.9%), TrueBeam (15 patients, 23.8%), Halcyon (four patients, 6.3%), and Elekta’s Infinity (five patients, 7.9%). (Note: The plans were created by a planning design mentor with over 20 years of radiotherapy experience using accelerator models that were employed during the patients’ actual clinical treatments). The dose limits for the targets and OARs were set according to the National Comprehensive Cancer Network (NCCN) guidelines (Ettinger et al., 2023) and the Radiation Therapy Oncology Group (RTOG) 0617 protocol (Bradley et al., 2015) as well as our hospital’s clinical requirements as follows:

• PTV: V60 Gy > = 95%, D0.03cc < 6,900 cGy (115% of prescription dose), conformity index (CI) >0.8

• Lungs: V30 Gy < 18%, V20 Gy < 28%, V5 Gy < 50%, D_mean < 12.5 Gy

• Heart: V30 Gy < 40%, D_mean < 25 Gy

• Spinal cord: D0.03cc < 45 Gy

For clinical plan evaluation, the variable “VxGy” is defined as the volume of the structure receiving a dose greater than or equal to x Gy, and “Dxcc” is defined as the maximum dose received by x cm3 of the structure. The maximum dose of the structure is typically expressed as D0.03cc. The equivalent uniform dose (EUD) function was used during optimization (Niemierko, 1997; Wu et al., 2002). Conformity was calculated with the Paddick conformity index (CI), calculated as follows: CI = (TV∗PIV)2/(TV∗PIV), where TV is the target volume and PIV is the total volume of the prescription isodose (Paddick, 2000).

The complexity of the plan was quantified with the the edge metric complexity (Younge et al., 2012), calculated as M=1MU∑i=1NMUi×yiAi, where the sum is over all control point apertures from i = 1 to N. Here, MU represents the total number of monitor units in the plan, MUi is the number of monitor units delivered through aperture i, Ai is the open area of aperture i, and yi is the aperture perimeter excluding the MLC leaf ends.

Automated plan design and optimization were performed with the RayStation 9A planning system and the Halcyon accelerator model using 6 MV X-rays in flattening filter-free (FFF) mode with a dose calculation grid of 2.5 mm × 2.5 mm × 2.5 mm and the collapse cone (CC) dose calculation algorithm. Three counterclockwise partial arcs and corresponding opposite clockwise partial arcs (182° 230°, 300° 60°, and 130° 178°) were used (see Fig. 1), with a dose rate of 800 MU/min; the collimator angle was set to 10° or 350° according to the target shape on beam eye view (BEV). Since the Halcyon accelerator supports the delivery of automated packed arcs, the use of multiple field did not increase the duration of the actual treatment process. Given that the manual and automated plans were created on different TPS versions and linac models, subsequent dosimetric comparisons should be interpreted in the context of platform-specific characteristics.

Figure 1 Beam arrangement for conventional lung cancer radiotherapy VMAT plans (clockwise arcs 182° 230°, 300° 60°, 130° 178° and their counterclockwise opposite arcs).

Principle of automatic progressive optimization

The DMPO algorithm used in the RayStation planning system involves two main steps: first, fluence optimization, followed by optimization of the MLC and other physical parameters of the field control points. The automatic progressive optimization process consists of three sequential phases:

Phase 1: Initialization. The system first assigns a set of optimization functions to each region of interest (ROI), including targets and organs at risk (OARs). Each function is defined by a type (e.g., MinDose, MaxEUD), a numerical dose level, and a weight. To allow flexible convergence during early iterations, the initial dose constraints are deliberately set with loose thresholds, as listed in Table 1.

Table 1 Initial optimization functions used in the automatic template for NSCLC plans with a prescription of 60 Gy.

Two auxiliary structures were created: ring = PTV expanded by 1.3 cm minus 0.3 cm, with MaxEUD (a = 150) to enhance target conformity and limit high-dose spillover; nt = Body minus PTV expanded by 1.3 cm, representing normal tissue outside the PTV, optimized using MaxEUD with a = 150 and 1 to constrain high and low doses, respectively. The Body DoseFallOff function enforces a dose gradient outside the PTV boundary, for example, reducing dose from 60 Gy to 0.95 × 60 Gy within 0.5 cm. The MaxEUD function optimizes equivalent uniform dose (EUD), and the DoseFallOff function is a gradient constraint function available in RayStation’s scripting interface. Here, sp0.3 indicates the spinal cord with a 0.3 cm expansion.

ROI	Optimization function	Weight	
PTV	MinDose 6000cGy	50	
CTV	MinEud 6000cGy with a = −150	50	
PTV	MinDvh 6000cGy to 98%	200	
PTV	MaxEud 6000*1.05 cGy with a = 150	50	
Ring	MaxEud 6000*0.95 cGy with a = 150	8	
nt	MaxEud 6000*0.80 cGy with a = 150	8	
nt	MaxEud 6000*0.1 cGy with a = 150	8	
sp0.3	MaxDose 4000 cGy	20	
sp0.3	MaxEud 3900 cGy with a = 150	10	
Heart	MaxEud 2200 cGy with a = 1	10	
Lungs	MaxDvh 500 cGy to 48%	20	
Lungs	MaxDvh 2000 cGy to 28%	20	
Lungs	MaxDvh 3000 cGy to 18%	20	
Lung-PTV	MaxEud 1200 cGy with a = 1	20	
Body	DoseFallOff from 6000 to 6000*0.95 cGy in 0.5 cm	20	
Body	DoseFallOff from 6000 to 6000*0.85 cGy in 1.0 cm	20	
Body	DoseFallOff from 6000 to 6000*0.75 cGy in 1.5 cm	20	
Body	DoseFallOff from 6000 to 6000*0.7 cGy in 2.0 cm	20	
Body	MaxDose 6000*1.1 cGy	200	

Phase 2: Iterative optimization and feedback evaluation. Typically, the number of iteration steps per cycle was set between 60 and 100. In each optimization cycle, the system performs a dose calculation and evaluates the feedback values for each assigned function:

• If a feedback value is below the threshold (set as 4 × 10−4 in our script), the dose objective for that function is adjusted—for example, by scaling the current dose value by 0.98—and the function is re-evaluated.

• These steps are repeated until the feedback values for all key ROIs (e.g., ring, nt, Body Dose-FallOff) exceed the threshold, indicating sufficient dose conformity and gradient steepness.

Phase 3: Convergence and plan complexity control. Once the optimization functions reach convergence, the process is terminated. To prevent overly complex plans, a control mechanism resets and restarts the entire optimization if more than three full iterations are executed without meeting the stopping criteria. This avoids excessive MLC segmentation complexity. All thresholds, iteration limits, and update factors used in this process are treated as tunable hyperparameters and can be adjusted according to institutional planning protocols and clinical experience.

The flowchart of the specific optimization process is as in Fig. 2.

Figure 2 Flowchart of the automatic progressive plan optimization.

The resetting of beams for more than three cyclic optimizations is to avoid creating an excessively complex plan.

During the iterative optimization process, at the end of each cycle and after normalization to the prescription dose, the system provides the current dose distribution and calculates the feedback value for each optimization function. The current dose values can be read by the code, and the dose target values for each optimization function can be adjusted. For example, the target value can be updated to the current dose value scaled by 0.98, and the feedback value can be recalculated. This updating process continues until the feedback value of the optimization function exceeds a preset threshold, such as 4e−4.

The iterative optimization ends when, after the cycle ends and the dose is normalized to the prescription dose, the feedback values of the optimization functions related to the ring, nt, and Body Dose-FallOff exceed the preset threshold (4e−4). This indicates that the optimization has yielded the desired dose gradient. Additionally, to prevent excessive iteration from leading to high plan complexity, if the number of optimization cycles exceeds 3, the optimization results are reset, and the plan is reoptimized.

Importantly, this optimization process design requires several hyperparameters, including the number of iterative optimizations, thresholds, and scaling factor for updating the dose target values. The selection of optimization function template also needs to be adjusted according to the requirements of the corresponding institutions. These hyperparameters are set on the basis of clinician experience but can also be flexibly adjusted to allow different institutions to apply this progressive automatic optimization script. The threshold for resetting optimization cycles (i.e., >3 iterations) was chosen based on clinical experience with the RayStation planning system, where excessive optimization tends to lead to more complex MLC segment shapes and increased treatment plan complexity. This threshold aligns with similar findings in other planning systems and reflects an empirically derived value aimed at balancing plan quality and clinical feasibility. The feedback threshold (4e−4) and reset iteration count (>3) were determined based on preliminary institutional validation using 60 test cases involving thoracic and pelvic radiotherapy. These values offered a robust trade-off between optimization convergence and plan complexity. We recommend that other centers adjust these parameters based on site-specific planning goals and equipment characteristics. Although the automated plans in this study were generated using RayStation 9.0 and the Halcyon accelerator, the scripting framework is compatible with other TPS versions and linac platforms. Therefore, any observed dosimetric differences may partially reflect differences in system configurations rather than optimization strategy alone.

Statistical analysis

According to the plan design steps, the dose to the target area in both groups was normalized such that 95% of the target volume received 100% of the prescription dose (60 Gy). Under these conditions, the dosimetric parameters of each plan were separately collected. Statistical analysis of the data was performed with SPSS 26.0 software. The normality of the distribution of the data normality was assessed with the Shapiro–Wilk method, the results of which revealed that the data for the manual plans and most of the automatic plans did not conform to a normal distribution (p value less than 0.05). Therefore, the paired-sample nonparametric Wilcoxon signed-rank test (α = 0.05, two-tailed) was used for comparisons between the groups.

Results

The CTV of the 63 NSCLC patients whose plans were included in this study was 169.39 ± 118.67 cm3, and the PTV was 286.8 ± 155.7 cm3. The total script runtime for automatic planning was approximately 25–35 min per case, including both structure preprocessing and optimization execution. Under the condition that 100% of the prescription dose (60 Gy) covers 95% of the PTV, we compared automated plans (AP) generated by the RayStation scripting workflow for Halcyon with manual plans (MP) used as clinical controls. Except for a slightly lower PTV_D98% value, the CTV and PTV dose levels in the APs were approximately 2% higher than those in the MPs, and the differences were statistically significant (p < 0.001). The conformity index (CI) of the PTVs in the APs was significantly higher than in the MPs, while the homogeneity index (HI) was slightly lower, as shown in Table 2. Figure 3 summarizes the statistical comparisons of target dose metrics.

Table 2 Dosimetric indicators of CTV and PTV for automated plans (AP) and manual plans (MP).

Variable	AP	MP	Difference	Percentage difference (%)	P value	
CTV_D98% (cGy)	6,091.3 ± 32.028	6,044.8 ± 41.74	46.5	0.76926	<0.001	
CTV_D2% (cGy)	6,561.3 ± 97.457	6,327.0 ± 74.898	234.3	3.7032	<0.001	
CTV_D50% (cGy)	6,344.5 ± 67.23	6,202.2 ± 57.023	142.3	2.2943	<0.001	
CTV_Dmean (cGy)	6,339.0 ± 62.993	6,197.3 ± 55.765	141.7	2.2865	<0.001	
PTV_D98% (cGy)	5,865.8 ± 52.797	5,919.0 ± 71.053	−53.2	−0.8988	<0.001	
PTV_D2% (cGy)	6,557.7 ± 91.531	6,324.4 ± 72.694	233.3	3.6889	<0.001	
PTV_D50% (cGy)	6,292.8 ± 52.022	6,177.3 ± 45.735	115.5	1.8697	<0.001	
PTV_Dmean (cGy)	6,277.9 ± 46.454	6,166.7 ± 42.044	111.2	1.8032	<0.001	
PTV_D0.03cc (cGy)	6,672.3 ± 109.54	6,381.2 ± 82.644	291.1	4.5618	<0.001	
PTV_HI	0.10984 ± 0.019464	0.065522 ± 0.019251	0.044318		<0.001	
PTV_CI	0.87429 ± 0.04248	0.83317 ± 0.061347	0.04112		<0.001	

Figure 3 Statistical analysis results for D2%, D50%, D98% and mean dose of CTV and PTV.

The evaluation of the OARs included various dosimetric indicators for the lungs, heart, and spinal cord. The lung volume was 3,177.5 ± 1,050.0 cm3, and the heart volume was 691.44 ±116.87 cm3. The results of the evaluation indicated that the following dose indicators in the AP group were significantly lower (p < 0.001) than those in the control group: lungs, V5Gy, V10Gy, V20Gy, V30Gy, and mean dose (Dmean); heart, V30Gy and Dmean; and spinal cord, maximum dose (D0.03cc). Among them:

• Lungs_Dmean: AP group, 1,002.2 ± 277.79 cGy; MP group, 1,108.0 ± 214.17 cGy; difference, −84.6 cGy (−7.6354%).

• Heart_Dmean: AP group, 741.07 ± 443.15 cGy; MP group, 881.84 ± 450.6 cGy; difference, −140.77 cGy (−15.963%).

• Spinal Cord D0.03cc: AP group, 3,379.3 ± 651.32 cGy; MP group, 3,623.2 ± 470.54 cGy; difference, −243.9 cGy (−6.7316%).

The specific results are presented in Table 3, and the statistical analysis results shown in Figs. 4 and 5.

Table 3 Dosimetric indicators of organs-at-risk for automated and manual plans.

Variable	AP	MP	Difference	Percentage difference (%)	P value	
Lungs_V5Gy (%)	35.618 ± 9.1372	39.181 ± 7.2571	−2.873	−7.2375	<0.001	
Lungs_V10Gy (%)	27.682 ± 7.4492	29.656 ± 5.8722	−1.436	−4.8422	<0.001	
Lungs_V20Gy (%)	18.293 ± 5.8844	20.197 ± 4.3913	−1.509	−7.4714	<0.001	
Lungs_V30Gy (%)	12.010 ± 4.5671	13.717 ± 3.4865	−1.412	−10.294	<0.001	
Lungs_Dmean (cGy)	1,002.2 ± 277.79	1,108.0 ± 214.17	−84.6	−7.6354	<0.001	
Heart_V30Gy (%)	7.4292 ± 6.0642	9.7447 ± 6.7578	−2.3155	−23.762	<0.001	
Heart_Dmean (cGy)	741.07 ± 443.15	881.84 ± 450.6	−140.77	−15.963	<0.001	
SpinalCord_D0.03cc (cGy)	3,379.3 ± 651.32	3,623.2 ± 470.54	−243.9	−6.7316	<0.001	

Figure 4 Dose volume analysis results for organs at risk (V5Gy, V10Gy, V20Gy, V30Gy of lungs, and V30Gy of heart).

Figure 5 Dose analysis results for organs at risk (mean dose of lungs, Lung-PTV, heart and D0.03cc of spinal cord).

The number of MUs for the Halcyon plans generated with the RayStation automatic progressive optimization script was 648.02±109.54 MU, with a plan complexity quantified using the edge metric (mean ± SD: 0.0677 ± 0.0093 mm−1), as defined by Younge et al. (2012) and Younge et al. (2016). In the manual plan control group, the number of MUs was 522.61 ± 124.69 MU, with a corresponding edge metric of 0.0743 ± 0.0148 mm−1. Both the MU and complexity differences between the two plan groups were statistically significant (p < 0.001). It should be noted that the observed differences between automated and manual plans may also be influenced by differences in TPS versions (RayStation 9.0 vs. 4.5) and linac platforms (Halcyon vs. multiple conventional linacs). As such, the findings reflect the performance under the studied configuration and should not be generalized as intrinsic advantages of the automation algorithm itself.

Discussion

In this study, radiotherapy plans for 63 NSCLC patients were analyzed to compare dosimetric and complexity characteristics between automated plans (APs), generated using a progressive scripting workflow in RayStation version 9.0 for Halcyon, and retrospectively collected manual clinical plans (MPs), created in RayStation version 4.5 on a range of linac platforms. As shown in Table 2 and Fig. 3, the APs achieved higher mean dose levels to the CTV and PTV (approximately 2% higher) with statistical significance (p < 0.001), although the PTV_D98% was slightly lower. Table 3 and Figs. 5 and 4 also indicate lower dosimetric values for the lungs, heart, and spinal cord in the AP group. Additionally, the APs showed lower monitor unit usage and edge-metric–based plan complexity. Figure 6 also illustrates this trade-off: the automated plans achieved reduced lung V20Gy and V5Gy while maintaining target coverage that met institutional clinical acceptability criteria. Specifically, each AP satisfied the requirement that 95% of the PTV received at least 60Gy, with a maximum dose not exceeding 115% of the prescription dose (i.e., PTV D0.03cc < 69Gy), as verified by experienced radiation oncologists based on evaluation guidelines adapted from RTOG 0617 (Bradley et al., 2015). These differences may result from a combination of scripting strategies—such as prioritizing dose conformity and steeper gradients—and differences in TPS versions and treatment delivery platforms. In particular, the AP script allows a maximum dose up to 110% of the prescription, thereby enabling greater flexibility in sparing surrounding tissues. This relaxed upper dose constraint facilitates the formation of steeper dose gradients outside the PTV boundary, improving the optimizer’s ability to reduce exposure to adjacent organs at risk. The reduced plan complexity in the AP group may also stem from the dual-layer stacked MLC design and the script’s constraint-aware dose modulation behavior. However, as the AP and MP plans were generated using different versions of the TPS and on different linac models, the observed differences may partly reflect variations in hardware capabilities and dose calculation algorithms. Therefore, these results should be interpreted within the context of the specific system configuration used in this study, and not as a direct comparison of planning quality between automation and manual methods.

Figure 6 Mean DVHs of 63 patients comparing automated plans (AP) and manual plans (MP) for the PTV, lungs, and heart.

APs demonstrate reduced OAR doses with minimal compromise in PTV coverage.

The Halcyon accelerator is capable of automatic couch shifting and daily image-guided radiation therapy (IGRT), allowing the isocenter of the radiation field to be freely selected. Therefore, in this study, we directly used the center point of the PTV as the isocenter for all the treatment arcs. In conventional lung cancer treatment planning, to protect healthy lung tissue, treatment arcs are selected with a preset uniform angle range (the collimator angle is selected as 10 or 350° on the basis of the BEV direction), thus eliminating plan quality inconsistencies caused by beam angle selection. By following the steps in this study, even physicists and dosimetrists with limited experience can create conventional lung cancer radiotherapy plans that meet clinical requirements, with a sufficient dose gradient and normal organ sparing.

In the current clinical radiotherapy process, treatment plans usually require multiple assessments and meticulous adjustments. This iterative process is not only time-consuming and labor-intensive but also highly dependent on the experience of the physicists and dosimetrists. Although KBP (Zhu et al., 2011; Appenzoller et al., 2012; Wu et al., 2016) and MCO (Fjellanger et al., 2023) technologies have addressed these issues to some extent, some limitations still persist in optimizing plans with these systems. Early versions of KBP relied on manually extracting features from historical plans, limiting complex data input and prediction accuracy. Advances in deep learning have led to new possibilities for dose distribution prediction, significantly improving the accuracy and consistency of plans by creating precise 3D dose distribution models (Ronneberger, Fischer & Brox, 2015; Shafiq & Gu, 2022; Milletari, Navab & Ahmadi, 2016). Many deep learning-based volumetric dose prediction models have also been applied in the treatment of lung tumors (Barragán-Montero et al., 2019; Shao et al., 2021).

Automatic planning enables consistent plan generation through standardized workflows and iterative optimization, potentially reducing human variability in manual planning. Prior studies have demonstrated its benefits across several tumor sites (Yang et al., 2020; Hirotaki et al., 2023). In our study, the automated plans achieved favorable dose distribution and conformity metrics under the current system configuration, which includes RayStation 9.0 and the Halcyon platform. While these findings align with previous literature (Hazell et al., 2016; Hansen et al., 2017; Hussein et al., 2016), it is important to note that platform-specific characteristics may also contribute to the observed differences. Additionally, the plans generated with the Halcyon accelerator model and the RayStation planning system’s progressive automatic optimization scheme demonstrated high plan quality and consistency. These plans can also be used as training materials for knowledge bases and dose prediction, facilitating further research.

This study used retrospectively collected clinical radiotherapy plans as the manual planning (MP) control group. While this design enabled comparison across a broad range of real-world clinical scenarios, it also introduced potential confounding due to the heterogeneity of planning systems and delivery platforms. In particular, the AP group was generated using RayStation 9.0 and the Halcyon platform, while the MP group included plans created in RayStation 4.5 across multiple linacs, such as Trilogy, TrueBeam, and Elekta Infinity. These system-level differences—especially in MLC architecture, beam characteristics, and algorithm performance—may have influenced the observed results. For example, the Halcyon accelerator employs a dual-layer, stacked MLC design that simplifies aperture modulation and may contribute to the lower plan complexity observed in the AP group. While the scripting strategy plays a role in shaping dose gradients, the impact of hardware design cannot be overlooked. Therefore, interpretations of plan complexity or dosimetric efficiency should consider the underlying system configurations. Future work should include intra-platform comparisons, such as AP versus MP plans both generated using Halcyon, to reduce inter-device variability.

This study has several limitations. First, differences in TPS versions and delivery platforms between AP and MP groups may confound direct comparisons, limiting the attribution of dosimetric differences to the optimization algorithm alone. Second, the sample size was relatively small (63 patients), and data were drawn from a single institution. Third, the analysis focused solely on NSCLC, and the findings may not generalize to other tumor types with different planning requirements. Fourth, because of the retrospective nature and the time span over which the MP data were collected, variations in equipment performance and protocol changes over time may also have affected results. Lastly, the observation period was short, and long-term clinical outcomes such as survival and radiation-induced toxicities were not evaluated. Future prospective, multicenter studies are needed to validate the clinical generalizability and long-term benefits of automated planning.

Although this study primarily focused on dosimetric parameters, the findings offer preliminary insights into differences between AP and MP in terms of PTV coverage, OAR protection, and dose gradients. However, the absence of a blinded clinical review limits the clinical interpretability of the results. Future studies should incorporate blinded assessments by experienced radiotherapy physicians to evaluate the clinical feasibility and acceptability of automated plans. In addition, correlating dosimetric benefits with clinical endpoints such as survival, toxicity, and treatment response will be critical in determining the real-world impact of automated optimization systems.

Lastly, although the data in this study were derived from retrospective cases collected several years ago, they still offer meaningful clinical insights into radiotherapy planning across evolving technological contexts. These data serve as a baseline for comparing AP and MP approaches and help contextualize observed outcomes in relation to earlier planning practices. To enhance the relevance and applicability of future research, we plan to incorporate recent patient cohorts and contemporary system configurations to evaluate the performance of automated planning under current clinical conditions.

Conclusion

In summary, this study explored the feasibility and dosimetric characteristics of automated radiotherapy planning using a progressive scripting algorithm in RayStation for the Halcyon platform. Under the studied configuration, automated plans demonstrated favorable target coverage, organ-at-risk sparing, and plan consistency compared to retrospectively collected manual plans. These findings suggest that automated planning has the potential to improve planning efficiency and standardization, particularly in settings with well-integrated hardware–software environments.

However, the observed advantages may partly reflect differences in treatment planning systems and delivery platforms, and should not be interpreted as intrinsic superiority of the automation algorithm. Further studies using matched TPS and linac configurations, along with clinical outcome validation, are needed to fully evaluate the generalizability and clinical impact of automated planning in routine practice. With continued technical refinement and clinical validation, automated planning could become an important component of personalized and efficient radiotherapy workflows.

Supplemental Information

Supplemental Information 1 Source code for automatic progressive optimization of treatment plans on RayStation

Supplemental Information 2 Raw dosimetric data of automatric planning (AP) collected in this study

Supplemental Information 3 Raw dosimetric data of manual planning (MP) collected in this study

The authors would like to thank AJE for English language editing services.

Additional Information and Declarations

Competing Interests

Author Contributions

Human Ethics

Data Availability

The authors declare there are no competing interests.

Kainan Shao conceived and designed the experiments, performed the experiments, authored or reviewed drafts of the article, and approved the final draft.

Fenglei Du conceived and designed the experiments, authored or reviewed drafts of the article, and approved the final draft.

Lingyun Qiu performed the experiments, authored or reviewed drafts of the article, and approved the final draft.

Yinghao Zhang performed the experiments, prepared figures and/or tables, and approved the final draft.

Yucheng Li analyzed the data, prepared figures and/or tables, and approved the final draft.

Jieni Ding analyzed the data, prepared figures and/or tables, and approved the final draft.

Wenming Zhan analyzed the data, prepared figures and/or tables, and approved the final draft.

Weijun Chen conceived and designed the experiments, authored or reviewed drafts of the article, and approved the final draft.

The following information was supplied relating to ethical approvals (i.e., approving body and any reference numbers):

The retrospective study was approved by the Medical Ethics Committee of Zhejiang Provincial People’s Hospital (No. QT2024085).

The following information was supplied regarding data availability:

The source code, raw dosimetric data of automatic planning and raw dosimetric data of manual planning are available in the Supplementary Files.

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
