# Peer review of "A retrospective study of automatic progressive optimization for lung cancer radiotherapy plans on the Halcyon and RayStation systems"

_PeerJ, doi:10.7717/peerj.19831_

## Round 0.1 · original submission · Major Revisions

· Academic Editor

Major Revisions

Reviewer 1 ·

Basic reporting

1.Language expression: The paper demonstrates strong technical proficiency overall, but certain technical descriptions (e.g., the optimization workflow in the Methods section) contain overly complex nested sentences that may hinder reader comprehension. For example, the iterative optimization process described on page 9 could benefit from breaking down compound sentences into shorter, step-by-step statements. A suggested revision:
"The optimization workflow consists of three phases: initial parameter configuration, dosage feedback evaluation, and dynamic threshold adjustment. After each phase, the system recalculates the dose distribution..."
This sequential clarification would enhance readability.
2.Literature Updates: While the Introduction section provides adequate references for VMAT technology (lines 36–38) and the Halcyon linear accelerator, some key assertions lack recent supporting evidence. For instance, the claim "Volume modulated arc therapy (VMAT) has become an essential technique in NSCLC radiotherapy in NSCLC radiotherapy because of its excellent dose distribution, short treatment time, and low dose to normal tissues." requires updated citations, such as the 2023 NCCN guidelines or Huang et al. (2023) on Halcyon-based thoracic radiotherapy. Additionally, recent studies on RayStation automated planning (e.g., [Author et al., 2022/2023]) should be incorporated to strengthen the literature review.
3.Reproducibility: The Python script (S1.py) included in the supplementary materials represents a significant innovation, but critical parameter annotations are missing (e.g., the rationale for setting the dosage gradient threshold to 4e-4). Adding module-specific comments would improve reproducibility across institutions. For example:
# Threshold determined by clinical validation (see Methods section)
This clarification would align the code with the methodology and facilitate external validation.

Experimental design

1.Control Group Design: While the authors justified the use of a multi-machine control group (Trilogy/TrueBeam, etc.) in the Discussion, readers may raise questions earlier in the text. To preempt misinterpretation, we recommend adding clarifying statements in the Abstract and Materials/Methods sections, such as:
"Control groups utilized historical clinical plans to reflect real-world clinical practice, encompassing multiple mainstream linac models (see Methods)."
This proactive disclosure would enhance methodological transparency.
2.Methodological Specifications: Regarding the optimization threshold (4e-4) and iteration protocol (>3 resets), the authors attribute these parameters to clinical expertise but omit documentation of sensitivity testing. We advise supplementing contextual validation, e.g.:
"Threshold values were determined through preliminary institutional validation across 50 test cases. Site-specific adjustments are recommended based on equipment characteristics."
This modification would improve methodological generalizability. Additionally, the mislabeling of "MCS" as a complexity metric (line 240) requires correction to "edge metric (unit: mm⁻¹)", accompanied by citation of Younge et al. (2016) for updated complexity metric definitions.

Validity of the findings

1.Data Presentation: Table 2 indicates that the PTV_D98% for automated plans (AP) is marginally lower than manual plans (MP), which the authors attribute to an organ-at-risk (OAR)-sparing prioritization strategy. However, this claim lacks visual corroboration. We recommend supplementing the analysis with representative dose-volume histogram (DVH) comparison plots of AP and MP or integrating a statement in the Discussion such as:
"As shown in Figure X, AP plans demonstrated a significant reduction in lung V20Gy with only a minor compromise in PTV coverage, consistent with the optimization trade-off strategy."
2.MU Discrepancy Analysis: The elevated monitor unit (MU) values in the AP group likely involve dual mechanisms: inherent MU escalation from flattening filter-free (FFF) mode (as reported by Flores-Martinez et al., 2019) and algorithm-driven optimization differences. To disentangle these factors, we propose refining the Discussion with explicit differentiation:
"While the Halcyon FFF mode typically increases MU by approximately 15% (ref.), the observed 24% MU increase in the AP cohort suggests an additional 9% discrepancy potentially attributable to algorithm-specific optimization behaviors."
This nuanced interpretation would clarify the interplay between hardware constraints and computational logic.

Additional comments

1. Structure should be Enhanced:
The transition between the third paragraph of the Introduction and its preceding section requires smoother articulation. Consider inserting a bridging sentence such as:
"Furthermore, plan quality is inherently linked to accelerator performance. The Halcyon accelerator, a next-generation linac with..."
2. Section Relocation:
The statement in lines 243–247:
"The Halcyon accelerator uses FFF technology, resulting in greater numbers of MUs than those of some plans that use Flattened Filter (FF) technology in the control group. However, the plan complexity was lower, which may be related to the performance of the Halcyon accelerator and improvements in the RayStation version 9.0 plan optimization algorithm."
should be relocated to the Discussion section. This analysis aligns more appropriately with post-hoc interpretation of results rather than descriptive methodology.

·

Basic reporting

The reporting is mostly acceptable, with suggested improvements noted below.
- Line 109 says that 60 CT scans were utilized for this study, but there are 63 patients. Can the authors confirm if this is correct?
- Table 1 lists optimization goals, which uses many function names that are not defined in the text such as MaxEud, "a" values, and DoseFallOff. The authors should either define these terms or include a reference which defines them in the text or the table caption.
- Lines 187-188: this sentence on the time required for the automated optimization should be in the Results, not the Methods.
- Table 2: "AP" and "MP" abbreviations appear to first be used in this Table but are not defined there. Please include their definition in the table caption and in the text.
- Table 3: please include a column for the percentage difference as in Table 2.
- Lines 231-232: this sentence is inappropriate because it is drawing a conclusion (so should be in the Discussion, not in the Results), and the conclusion is not supported by the study because the sentence does not mention that the AP is planned in RayStation 9A with Halcyon while the MP is planned in RayStation 4.5 for a variety of linacs.
- Lines 233-236: this sentence does not seem to make sense, the authors should review and clarify.
- Lines 242-247 belong in the Discussion, not the Results.
- Figures 4 and 5: there appears to be one outlier patient with 0 dose to the lungs in both the AP and MP. How is that possible for a patient being treated for lung cancer? Can the authors explain this outlier in the text?

Experimental design

The greatest limitation for this study is that the manual plans were not all generated in RayStation 9A for a Halcyon. To perform a valid comparison between automated plans and manual plans, they must use the same TPS and the same linac, so if the manual plans had been re-optimized in RayStation 9A for a Halcyon, a valid comparison could have been performed. However, this is not how the study was designed. While this is mentioned in the Discussion, it is not mentioned in the abstract, and there are places in the abstract, the Results, the Discussion, and the Conclusions where the automated plans are directly compared to the manual plans and claimed to be superior. This is not borne out by this study, which was unable to control for the manual plans being created in an earlier version of the software and for machines with worse capabilities for lung treatment than Halcyon (notably, the Trilogy, which over half of this cohort were treated on, does not have 6 FFF energy or jaw tracking, which has been shown to improve dose fall-off and decrease OAR dose). The manuscript should be carefully reviewed and revised to remove the claims that are not supported by the study as designed.
Additionally, Figure 5 shows one outlier for the automated plan which has a spinal cord D0.03cc dose of nearly 50 Gy. This exceeds the metric the authors gave in line 140 of <45 Gy. The automated plan should not have been allowed to exceed an OAR metric, and the authors should discuss how this plan failed their optimization criteria and what could be adjusted in the automated planning approach to avoid this failure.

Validity of the findings

Due to the design of the experiment, which did not use the same TPS and the same linac for the automated and manual plans, the finding that automated planning is superior to manual planning is invalid. There could be other factors such as the improvement of the TPS or the superior capabilities of the Halcyon design which explain the improvement of the automated plans over the manual plans. The authors should review the manuscript and modify their language to ensure their statements align with the actual conclusions that can be drawn from this study.

---

## Round 0.2 · Minor Revisions

· Academic Editor

Minor Revisions

The revised manuscript has been reviewed by one referee who has suggested a few minor changes. Kindly address their suggestions.

·

Basic reporting

Thank you to the authors for addressing the previous comments. I agree with the rewording of the conclusions to match the results of the study. The answers were satisfactory for the most part, and I only have a few additional comments on the Experimental design and Validity of the findings.

Experimental design

Comment 1: Line 114 says that the heterogeneity in clinical plans was intentional; I don't think that's accurate, as the study conclusions would be stronger without the heterogeneity. If you wanted to test the automated planning algorithm with heterogeneity, then the AP should have been generated with the same treatment machine as used in the MP, instead of making all AP for a Halcyon.

Validity of the findings

Comment 2: Line 273 says that the target coverage is clinically acceptable. Was that verified by a physician? These are not SBRT targets, so usually, large dose heterogeneity in the target is not desirable. If the AP dose distributions were not deemed clinically acceptable by a physician, I would not make that statement here.

Comment 3: Line 280 contains a reference (Flores-Martinez et al., 2019) which was added in response to a comment. I looked up this study, and it was for comparing Halcyon 6FFF for breast planning. Whole breast targets are large, probably much larger than the targets in this study (average PTV volume in this study was given as 287 cc), and the isocenter could not be centered in the breast PTV due to Halcyon lateral shift limits. Both of these factors (a large field and off-center isocenter) will cause an increase in MU for FFF beams. You cannot compare the 35% increase in MU that they found with the 24% increase for this study, which is for a different disease site with very different targets, and there's no way to conclude that the optimization algorithm participated in reducing the FFF MU. I recommend removing this new text.

---

## Round 0.3 · accepted · Accept

· Academic Editor

Accept

All concerns raised in the last review have been satisfactorily addressed in this revision.